# A critical realist analysis of nursing educators' willingness to learn and teach patient safety in Sri Lanka: Study protocol

D. M. A. P. Dissanayake📍[1]*, S. S. P. Warnakulasuriya[2]

**1** Department of Nursing and Midwifery, Faculty of Allied Health Sciences, University of Sri Jayewardenepura, Nugegoda, Sri Lanka, **2** Faculty of Nursing, University of Colombo, Colombo, Sri Lanka

* ashoka_dissanayake@sjp.ac.lk

## Abstract

### Introduction

Nursing educators' willingness to learn and teach patient safety is identified as a challenge in the process of implementing patient safety education in nursing programmes. However, the literature lacks an in-depth analysis of the factors affecting their willingness. This study aims to identify factors affecting willingness of nursing educators to learn and teach new patient safety concepts.

### Methods and analysis

The study will be conducted in eight state universities, two private universities approved by the University Grants Commission, and Sixteen (16) Nurses Training Schools in Sri Lanka. The sample will include permanent nursing educators fluent in Sinhala. The research design adopts the Critical Realist Mixed Method Sequential Explanatory Design (CRMMSED), utilizing both qualitative and quantitative approaches. Data collection will occur in four phases; curriculum content verification, individual innovativeness scale validation, quantitative data collection through surveys, and qualitative data collection via semi-structured interviews. The data will be analyzed using SPSS and AMOS for quantitative data, and NVivo for deductive thematic analysis.

### Discussion

This study will provide valuable insights into nursing educators' perceptions of patient safety education in Sri Lanka and the integration of global patient safety competencies into local curricula. The findings will guide improvements in nursing education, addressing educators' concerns and enhancing healthcare outcomes. While focused on Sri Lanka, the studys' insights have broader implications for global nursing

**Data availability statement:** Deidentified research data will be made publicly available when the study is completed and published.

**Funding:** DMAP received a research grant from University Grants Commission- Sri Lanka, UGC/VC/DRIC/PG2018(II)/SJP/01. https://www.ugc.ac.lk/index.php?option=com_content&view=article&id=2348&Itemid=134&lang=en. The funders did not and will not have a role in study design, data collection and analysis, decision to publish, or preparation of the manuscript.

**Competing interests:** The authors have declared that no competing interests exist.

education. Additionally, the use of critical realist methodology will also contribute to the ongoing dialogue about its application in healthcare education research.

## Introduction

Patient safety (PS) has been identified as an emerging trend in health services, and the World Health Organization (WHO) advocates for its inclusion in education accreditation [1]. According to the WHO, PS refers to a framework of organized activities that promote cultures, processes, procedures, behaviours, technologies, and environments in healthcare that consistently and sustainably lower risks, reduce the occurrence of avoidable harm, make errors less likely, and reduce their impact when they do occur [2]. In the context of ensuring patient safety, the emphasis has been shifted from assigning individual blame to recognizing and addressing the impact of human factors and fallibility. It is essential that these new concepts are effectively transmitted to the next generation of healthcare professionals.

Education on PS has been identified as a proactive step towards safer care and there is a global agreement on PS in undergraduate healthcare education [3,4]. Numerous PS frameworks have been developed and integrated into nursing programmes worldwide, such as the Quality and Safety Education for Nurses (QSEN) in the USA [5], Australian National Patient Safety Education Framework [6], South Korean Patient Safety Competency Framework (PSCF) [7] and Learning Outcomes for Patient Safety in Undergraduate Nursing Curricula in Canada [8]. WHO-Multi Professional Patient Safety Curriculum is also based on the Australian framework.

However, curriculum implementers have reported the challenging nature of integrating PS into curricula. Limited understanding of the PS concepts [9,10] and reducing PS practices to just one aspect, such as focusing solely on reducing medication errors [11], can hinder its overall impact. Furthermore, accommodating PS into an already saturated nursing curriculum [12,13],and not recognizing its importance in clinical practice [4,11,14] pose additional challenges.

One of the prominent challenges in integrating PS into nursing curricula is the willingness of faculty to learn and teach these concepts [4,15,16]. Several other issues related to faculty, such as a lack of recognition of the importance of PS by the faculty and a lack of confidence in teaching these concepts, have also been identified as challenging [4,17,18]. Nursing educators must be willing to integrate and teach new PS concepts to produce nurses who can champion PS. Therefore, further research is needed to evaluate both the curricula and nurse educators' willingness to learn and teach these concepts.

### Patient safety education in the South Asian region and Sri Lanka

A few studies [9,19,20] have been conducted in the Southeast Asian region to assess the perceptions of PS, barriers to implementing PS and strategies to overcome these barriers. Pelzang et al., [9] have emphasized the need to integrate and provide PS training and education to all categories of healthcare professionals. The lack of resources has been identified as the main barrier to promoting patient safety and quality practices in India [19,20].

In 2016 adverse events reporting forms were introduced to state-sector hospitals in Sri Lanka. In 2019 of the 10324 adverse events reported, 30.46% were falls [21]. It is apparent that non-clinical events are reported more frequently. It can only be predicted that the actual incident numbers are higher [21,22]. A study done in a teaching hospital in Sri Lanka revealed that nursing officers who are competent in skills related to adverse events and, near misses were 6.35% [23]. Nearly half (46.4%) of student nurses have agreed that making errors is inevitable while a considerable proportion (38%) agreed on blaming and punishing people who commit errors [24]. These findings emphasise the dire need for improving PS education for nurses in Sri Lanka, particularly adverse events reporting.

The "WHO South East Asia Regional Strategy for patient safety (2016-2025)" has proposed for integrating PS principles and practices into all professional healthcare courses, including nursing [25]. This implies that Sri Lanka has strategically agreed to incorporate PS education in undergraduate nursing education. These new perspectives and obligations make PS education an innovative and emerging necessity, particularly for a developing country like Sri Lanka.

World Health Organizations' Multi-Professional Patient Safety Curriculum (WHO-MPSC) [26] provides a common foundation for PS education. The curriculum contains a wide array of topics on patient safety, human factors, complex systems, teamwork, learning from error, clinical risk management, quality improvements, engaging patients and carers, infection control, surgical safety and medication safety. "It contains information for all levels of faculty and staff, and it lays the foundation for capacity building in essential patient safety principles and concepts" [27]. The guide has been written to be applicable to different cultures and contexts using easily-understood language [27]. Ginsberg et al. [12] have found that low and middle-income countries are slow to progress towards integrating the aforesaid curriculum. They have examined the potential barriers pertaining to; a) curriculum content, b) context and c) implementation process. Insufficient training to enable faculty, lack of faculty enthusiasm/meaningful participation and lack of faculty cooperation to address implementation challenges were among the top three potential barriers to the implementation process. Therefore, the concerns of nursing faculty should be identified and addressed to ensure seamless integration of PS concepts into existing curricula [28].

### Factors that affect willingness to learn and teach a new concept

**1. Individual factors.**

**Stages of concern:** The Concerns-Based Adoption Model (CBAM) [29] describes how teachers' concerns over the acceptance and adoption of educational change vary. Seven Stages of Concern (SoC) about an educational innovation have been identified, in which individuals progress from implementing an innovation to becoming competent in using it (Table 1).

A 35-item Stages of Concern questionnaire is available to determine where someone is in the Stages of Concern range [29]. Although it may appear that progression through the stages of concern is a linear process, it is important to note that acceptance and adoption of an innovation are highly influenced by the intrinsic factors of an individual. Even as individuals

**Table 1. Typical expressions of concern about an innovation [29].**

| Stages of concern | | | Expressions of concern |
|---|---|---|---|
| "Impact" | 6 | Refocusing | I have some ideas about something that would work even better. |
| | 5 | Collaboration | I would like to coordinate my efforts with others, to maximize the innovations' effect. |
| | 4 | Consequence | How is my use affecting my students? |
| "Task" | 3 | Personal | I seem to be spending all my time getting my materials ready. |
| "Self" | 2 | Management | How will using it affect me? |
| | 1 | Informational | I would like to know more about it. |
| "Unconcerned" | 0 | Unconcerned | I am not concerned about it. |

gain more knowledge and experience with innovation, there is no guarantee that their earlier concerns will be resolved [29].

The authors of "Measuring implementation in schools: the stages of concern questionnaire", George et al. [29] noted that the development of higher-level concerns depends on the **individual's perceptions**, **the environmental context** and **the innovation** itself. Buabeng-Andoh [30] identified personal, institutional, and technological factors that encourage the use of computer technology in teaching and learning, such as ICT competence, computer self-efficacy, professional development, and technical support. Lai and Chen [31] also suggested similar factors, including individual characteristics such as perceived usefulness and ease of use, school characteristics such as incentives and support, and environmental characteristics such as supervisor and peer influence.

**Individual innovativeness:** When introducing an innovative idea, the process of adapting that idea is rarely linear [32]. Roger's diffusion of innovation theory explains that an individual's adoption of an innovation progresses through five stages: knowledge, persuasion, decision, implementation, and confirmation [32]. Additionally, he suggests that individuals will adopt innovations at different time frames depending on the adopter category to which they belong (Innovators, Early Adopters, Early Majority, Late Majority, and Laggards) [32].

The general innovativeness of an individual is typically viewed as a psychological construct or individual characteristic. It shapes an individual's disposition toward newness regardless of the kind of innovation [33]. Hurt et al., [34] define Individual Innovativeness as the "willingness of an individual to change". Previous studies have found a positive relationship between individual innovativeness and education technology adoption behaviour among academicians and education administrators [35,36].

Apart from Individual innovativeness, there are studies done to identify how personal characteristics such as age, gender, and education affect innovation adoption [37–39].

**2. Environmental factors.** Nursing educators' working environment includes the teaching facility and the hospitals where clinical teaching and practical exams are carried out. Previous studies in nursing have frequently utilized [40–42] Kanter's 'Theory of Structural Power in Organizations' [43] to examine structural empowerment. Structural empowerment refers to workplace conditions and policies that facilitate four dimensions. These dimensions are access to information, support from colleagues, resources such as time and technology, and opportunities for professional development and growth. These dimensions influence the nursing educators' work environment and their ability to provide high-quality education to students. Research done in the management and IT industries has found that positive working environments such as strong managerial support, and allocation of sufficient resources and incentives become crucial catalysts in embracing innovation [39,44]. As such, it could be predicted that nursing educators' environment could affect education innovation adoptions.

**3. Factors related to the innovation.** Perceived attributes (PA) are important factors to consider when studying the adoption of an innovation. Relative advantage, compatibility, complexity, observability, and trialability are the five main perceived attributes [32]. Relative advantage means how much better an innovation is perceived to be, compared to the idea it replaces. Compatibility refers to how well the innovation fits with the values, experiences, and needs of potential adopters. Complexity explains how difficult the innovation is to understand and use. Observability refers to how visible the results of the innovation are to others. Trialability refers to the ability of individuals to experiment with the innovation before fully adopting it. These perceived attributes have been used in previous research to investigate their effect on the concern of users when adopting an innovation [30,31].

### The methodological approach

The methodology is concerned with the theoretical and philosophical assumptions about methods [45]. It provides the frame of reference for the research which is influenced by the paradigm in which our theoretical perspective is placed or developed [46]. Another definition is "the specific theoretical level at which research methods and approaches are

conceived, chosen and justified" [47]. Choosing a set of methods as opposed to others can be called methodological justification of the choice of methods [47]. Methodology can be objectivist, subjectivist or critical realist.

**Objectivist methodology.** Objectivist methodology is based on positivism. It claims that reality exists independent of the human mind, and it can be understood only through objective methods of inquiry [48]. Objective research methods require quantifiable data about the world to represent reality. When used in social research, objectivist methodology faces debilitating limitations. It may oversimplify complex social phenomena by reducing them to quantifiable variables, potentially overlooking important nuances and contextual factors. This approach may not capture the full richness and complexity of human experiences and social interactions [47].

**Subjectivist methodology.** Subjectivist methodology is based on interpretivism claims that the social and human world is a human creation based on their subjective encounters with them. There is no objective reality. It emphasizes that research should aim at understanding how people make sense of their own world [47].The limitations of this methodology include relying heavily on individual interpretations, which are influenced by personal biases or social contexts [48]. The subjectivist methodology may not provide a comprehensive understanding of complex phenomena as it primarily focuses on individual perspectives and experiences, potentially overlooking broader structural or systemic factors [47].

**Critical realistic methodology.** Critical Realistic methodology tries to transcend the subjectivist and objectivist dichotomy in epistemology and methodology. The triumvirate of critical realism is ontological realism, epistemic relativism and judgemental rationality [49].

**Ontological realism:** CR posits that reality is intransitive. There is an objective reality in the social world outside the human mind regardless of our ability to understand or decode it. Social reality includes languages, social processes, classes, economies, religion, whereas natural reality includes plants, trees, gravity etc. [49,50].

**Epistemic relativism:** Human knowledge gets limited by the methods we would use to uncover reality. Therefore, human knowledge is transitive. Our knowledge is finite, contextual and fallible [49,51,52].

**Judgemental rationality:** Critical realists should evaluate diverse and competing claims about the world before making a judgement. Human knowledge is not just facts or opinions but essentially a rational judgement [49].

Critical realism posits that the world is a stratified open system where structures and mechanisms with causative powers (real domain) generate events, which occur regardless of whether we experience them or not. It suggests that what happens in the world (actual events/Actual domain) is not the same as what is observed/experienced (empirical phenomena/empirical domain). Therefore, critical realism recognizes that we cannot reduce actual events to empirical phenomena or causative mechanisms to empirical phenomena. Critical realism is a meta-theory. As a meta-theory, critical realism is more an ontology than methodology. It focuses primarily on the nature of reality and does not firmly prescribe how to capture or know that reality [52].

Critical realism is not limited to a specific research method or approach, but instead provides a framework for understanding the underlying reality of the phenomena being studied. This allows researchers to use various methods to study different aspects of the same phenomenon and to integrate findings from different methods to gain a more comprehensive understanding. The focus is on understanding the mechanisms and structures that generate the observed phenomena, rather than adhering to a particular methodology or technique. This flexibility in approach makes critical realism a valuable tool for researchers in a variety of fields who seek to gain a deeper understanding of complex phenomena [50].

In this study, researchers will utilize critical realism's depth reality [53] to overcome the limitations of the empirical nature of the approaches used to examine the stages of concern in earlier studies [54,55]. In this study empirical evidence will be obtained on the willingness to learn and teach PS, across individual, environmental and innovation-related areas. Critical Realism's depth reality will be used to understand complexities that are not obvious and visible (Fig 1).

Critical realists use double hermeneutics, abduction and retroduction to conceptualize the actual events and generative mechanisms. Studying objects of nature involves simple hermeneutics while studying social objects involves 'double hermeneutics' as the social scientists are 'interpreting other people's interpretations' [50]. It is essential to distinguish what

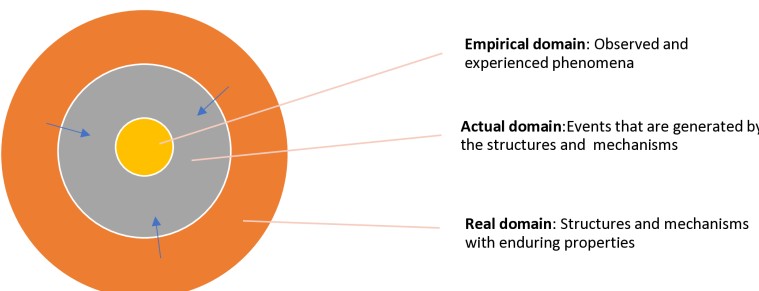

**Fig 1. The stratified ontology of critical realism.**

needs to be explained and explanations given by others. Charles S. Peirce was the first to report abductive inference in 1932 [56]. It is the logical method of discovering or drawing conclusions from, circumstances and structures that are not given in individual empirical data. Retroduction is the mode of inference in which events are explained by postulating (and identifying) mechanisms which are capable of producing them [53,57]. This allows the researcher to go beyond empirical findings and critically examine the in-depth reality of a phenomenon.

## Aim of the study

To conduct a critical realist analysis of factors influencing willingness to learn and teach adverse events related PS concepts by nursing educators.

## Significance of the study

There is very limited understanding on the current level of patient safety education in Sri Lanka. The results of this study will generate new knowledge on how PS has been perceived by Sri Lankan nursing educators. It will also provide an understanding of the possibilities of integrating internationally recognized PS competencies into local nursing curricula in Sri Lanka. Critically analysed willingness of nursing educators will contribute to equip education institutions with information on individual, environmental, and PS content-related changes and improvements. In addition, this study will enable curriculum developers to proactively address the concerns of nursing educators, leading to better outcomes. Improved education on PS can minimize the effects of adverse events in healthcare and improve the public image of healthcare institutions. Even though the study will not generate a generalizable theory, the critical realist methodology used in this study will be useful for the future researchers, when conducting similar studies in any part of the world.

## The conceptual framework

The conceptual framework of this study (Fig 2) was developed to graphically present the theoretical and methodological underpinnings of the study. Anticipated events, structures and mechanisms were included acknowledging the inevitable alterations, post-data analysis.

## Objectives

### General objectives.

1. To identify and examine the factors that influence nursing educators' willingness to learn and teach adverse events related patient safety concepts.

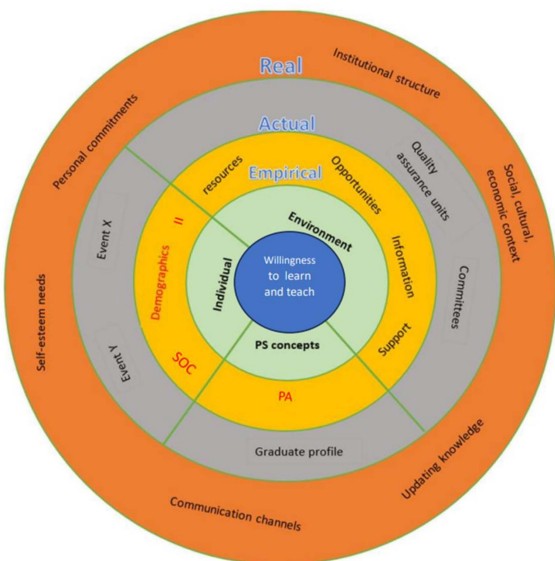

**Fig 2. The conceptual framework.**

2. To analyse the underlying structures and generative mechanisms that contribute to the development of variations in willingness to learn and teach patient safety, among nursing educators.

**Specific objectives.**

1. To assess the extent of inclusion of adverse events related patient safety concepts in the curricula of selected nursing programmes.

2. To translate and validate the Sinhala version of the Hurt-Joseph-Cook individual innovativeness questionnaire for use with nursing educators.

3. To evaluate the level of individual innovativeness among nursing educators.

4. To determine the specific stages of concern exhibited by nursing educators regarding the integration of adverse events related patient safety concepts.

5. To explore the correlations between Individual Innovativeness and the stages of concern among nursing educators.

6. To examine the perceived attributes and environmental factors that facilitate or hinder the adoption of adverse events related patient safety concepts by nursing educators.

7. To synthesis the actual events, generative mechanisms and structures that contribute to the facilitation or hindrance of the adoption of adverse events related patient safety concepts.

## Materials and methods

In Critical Realistic methods, quantitative and qualitative data collection is frequently termed as extensive methods and intensive methods, respectively [50,57]. In this study, empirical data collected through extensive methods will be used to explain the phenomenon, while abductive inferencing will be used to identify actual events which are not explicitly linked to the phenomenon. Retroduction will be used to identify structures and generative mechanisms in the real domain.

## Study design

Mixed Method Sequential Explanatory (MMSE) [58] design and the staged model for explanatory social sciences (DAARCC) proposed by Danermark (2019), were adopted when designing the research study. DAARCC stands for D- description, A -analytical resolution, A- abduction/theoretical redescription, R- retroduction, C-comparison between different theories and abstractions, C-concretization and contextualization [50]. Baisley [59], has proposed a Critical Realist Mixed-Methods Sequential Explanatory Design (CRMMSED), a research design combining critical realistic concepts, the staged model for explanatory social sciences and the original mixed-method sequential explanatory design. This research design follows the same design as CRMMSED: extensive data collection and analysis, linking phase, intensive data collection and analysis phase and interpretation phase. Whereas description (D), analytical resolution (A), and abduction/theoretical redescription (A) stages of the DAARCC model recur during conceptualization and data collection (Table 2).

## Study setting

The study will be carried out in Nursing Diploma and B.Sc. Nursing programmes that are conducted in state universities (n = 8), private universities approved by the University Grants Commission, Sri Lanka (n = 2) and Nurses Training Schools (NTS) attached to general and teaching hospitals (n = 16) under the Ministry of Health, Sri Lanka.

## Study population

Permanent nursing educators ($N_{total}$ = 413) who are capable of reading, speaking and comprehending the Sinhala language ($N_S \approx$ 378).

## Inclusion and exclusion criteria

**Inclusion criteria.**  Permanent academic staff members whose basic qualification is nursing will be included in the study. Participant should teach at least one core nursing subject (ex. Fundamentals of Nursing, Clinical Nursing subjects, Management in Nursing, etc.) and capable of reading, speaking and comprehending Sinhala language.

**Exclusion criteria.**  Academic staff members who are on maternity leave, overseas study leave and overseas sabbatical leave.

Table 2. Comparison of study designs.

| Sequence in this study | | Corresponding CRMMSED phases | Corresponding MMSE phases | Corresponding DAARCC stages |
|---|---|---|---|---|
| Conceptualization | | | | D - Description, A - Analytical resolution, A - Abduction/theoretical redescription |
| Data collection and analysis | **Phase 1 -** extensive data collection and analysis | 1. Extensive data collection and analysis | • Quantitative data collection <br> • Quantitative data analysis | D - Description, A - Analytical resolution, A - Abduction/theoretical redescription |
| | **Phase 2 -** Linking phase | 2. Linking phase | • Connecting quantitative and qualitative phases | |
| | **Phase 3 -** Intensive data collection and analysis | 3. Intensive data collection and analysis | • Qualitative data collection <br> • Qualitative data analysis | |
| **Phase 4 -** Interpretation | | 4. Interpretation phase | • Triangulation of the qualitative and quantitative results | R - Retroduction, C - Comparison between different theories and abstractions, C - Concretization and contextualization |

## Sampling technique

The sampling technique for this study involves purposive sampling and stratified random sampling. In **phase 1**, all available curricula in selected institutions will be included in content verification using a purposive sampling technique.

For the self-administered questionnaire, two sets (sample 1 and sample 2) of nursing academics will be recruited. The first set (sample 1) will be used to perform exploratory factor analysis (EFA) of II Scale and will be selected using stratified random sampling. Data from sample 2 will be used to questionnaire survey analysis.

The sample size will be determined based on the rule of thumb of 5–10 subjects per item in instrument validation studies [60] resulting in a sample size of 105 (20 items in II scale, 5:1 subject to variable ratio and adding 5% non-response rate). The participants will be selected from three types of institutions; state universities (SU), private universities (PU), and training schools (NTS) using stratified random sampling.

The second set (sample 2) will be used for the confirmatory factor analysis (CFA). This sample will consist of 185 participants who were selected using stratified random sampling and not included in sample 1.

## Sample calculation

Total population of nursing academics are 413 (SU = 99, PU = 17, NTS = 297). To leverage exclusions due to Sinhala language proficiency 35 (SU = 10, PU = 5, NTS = 20) were exempted. Approximately 378 were left as the population of nursing educators fluent in Sinhala.

Sample 1 ($n_1$) for EFA of II Scale (20 items; 1 item: 5 educators).

$$n1 = 100 + \textit{non respondents } 5\% = 105.$$

Subsampling of $n_1$ using stratified random method:

- $n_1$ (SU) = 89/378 X 105 =25.

- $n_1$ (PU) = 12/378 X 105 =3.

- $n_1$ (NTS) = 277/378 X 105 =77.

$$\textit{Remaining population for sample } 2 \; (\textit{n2}) \; \textit{calculation} = 378 - 105 = 273.$$

Sample 2 for CFA and questionnaire survey.

Calculation 1: Using Yamane's formula [61,62] n = N/ (1+N(e)*2), for a population of 273 with a margin of error of 5%, the sample is calculated to be 162. A sample size of 170 results after adding 5% non-respondents.

Calculation 2: Considering factor analysis for the SoC questionnaire, as subsequent validation is recommended by Tsang et al [63], 175 (35 items; 1 item: 5 participants) are required to calculate CFA. A sample size will be 184 results after adding 5% as non-respondents for CFA of SoC questionnaire.

Since calculation 2 results a higher number 184 ($n_2$) was selected as the sample size.
Subsampling of $n_2$ using stratified random method:

- $n_2$ (SU) = (89-25)/273 X 184 =43.

- $n_2$ (PU) = (12-3)/273 X 184 =6.

- $n_2$ (NTS) = (277-77)/273 X 184 =135.

In **phase 2**, semi-structured interviews will be conducted with nursing educators until data saturation is reached. Participants will be selected by purposive sampling method.

### Study instruments

**Self-administered questionnaire.** There will be an online questionnaire consisting of three (3) parts; personal characteristics, Stages of Concern Questionnaire (SoC) questionnaire and Individual Innovativeness (II) scale.

**Personal characteristics:** Details on gender, age, religion, ethnicity, personal income, marital status, education and employment (university, academic/ admin grade, years of teaching experience, years of clinical experience) will be ascertained in this part.

**Stages of Concern Questionnaire (SoC):** The original SoC had been tested for reliability (test/retest reliability ranges from 0.65–0.68) and validity (alpha-coefficients range from 0.64–0.83) [29,64]. SoC's Sinhala and Tamil translations (35 items) have been validated in Sri Lanka, and Cronbach alpha of the Sinhala version (mean) was reported to be 0.78 indicating good internal consistency. Content validity has been evaluated by ten (10) language and content experts [65]. Psychometric properties of a Turkish translation has been shown to have weak internal consistency and model fit [66]. The reliability for four of the seven stages was low (<0.70), and the inter-correlational matrix did not provide support for the hypothesized relationships among the stages. A shortened (16-item) version has been suggested to increase reliability (0.66–0.82) and validity. A validation study in Hong Kong [67], has found that all the fit indices were not satisfactory and the correlations among some factors were too high. They have suggested a 5-stage model with 22 items. The item-total correlations of the 22 items ranged from 0.46 to 0.70 and the alphas of the five scales varied from 0.75 to 0.84. During this study construct validity of all 3 models will be evaluated via Confirmatory Factor Analysis (CFA) to identify the best model fit in Sri Lankan context.

The validated SoC questionnaire will be used along with the change of "name of the innovation" from "5E instructional method" to "adverse events related patient safety concepts (AER-PS)".

**Individual Innovativeness (II) scale:** This scale has been developed to measure an individual's orientation towards innovation [34]. This has shown strong psychometric characteristics; Cronbach α above 0.80 and it has been used as a valid measure of general innovativeness in earlier studies [68–70]. A study done in Pakistan found reliability to be above 0.75 and values of model fit were close to 0.90 [71].

The scale has 20 items and it ranges from options Strongly Disagree = 1; Disagree = 2; Neutral = 3; Agree = 4; Strongly Agree = 5 respectively to rate each item.

- Scores above 80 are classified as Innovators.

- Scores between 69 and 80 are classified as Early Adopters.

- Scores between 57 and 68 are classified as Early Majority.

- Scores between 46 and 56 are classified as Late Majority.

- Scores below 46 are classified as Laggards/Traditionalists.

Translation of II scale to Sinhala language, cross-cultural adaptation and psychometric analysis will be done following the established guidelines on cross-cultural adaptation [72] and psychometric analysis [73,74].

### Data collection

Data collection is not anonymised. In order to protect privacy and confidentiality; names and university affiliations will be replaced with codes during data processing. Any data requested during publications will also be anonymised. Data will be stored in password-protected documents on a secure Google Drive linked to the researcher's faculty email address and later data will be transferred to an external storage device. After 10 years from the submission of the thesis, the data will be deleted. Only the investigator and supervisor will have access to non-anonymised data.

**Phase 1: Extensive/ quantitative procedures.**

**Curriculum content verification:** This will be used as a way of fact-checking/validating the adverse events related (AER) content of PS is being taught. Soft copies of the curricula will be requested from deans of the faculties and the Director-Nursing (Education), Ministry of Health, Sri Lanka. Upon receiving, a word search of the relevant PS content will be carried out. A Microsoft Excel matrix will be used to document which AER-PS concepts are mentioned in curricula.

**Self-administered questionnaire:** Authorization will be obtained from the relevant deans of faculties and training schools' principles to collect data from their academic staff members and to verify the curricula content. A random sample of nursing educators from each institution category (state universities, private universities and training schools) will be selected. An online questionnaire link will be emailed or texted to selected nursing educators. The selection of participants will not be based on ethnicity. An information sheet will be provided to participants and will contain a brief description of the study, ethical approval, details of investigators, the procedure for asking questions, and participants' rights. Prior to accessing the questionnaire, a consent form will be provided for participants to acknowledge their voluntary participation and the option to withdraw their consent at any time. Non-respondents will receive a reminder on the following week and a phone reminder on the 2nd week.

**Phase 3: Intensive/ qualitative procedures. Semi-structured interviews:** Interviews will be conducted to identify perceived attributes of selected PS concepts and environmental attributes which facilitate or inhibit the adoption of selected concepts. One-to-one interviewing of the selected participants will be carried out in person or virtually according to the participant's preference. If done in person, it will be at the participants' academic institution. An interview guide developed by the principal investigator will be used (S1 File). Participants will be requested to go through the information sheet and provide consent form prior the interview. Interview will be conducted in participants preferred language (Sinhala, Tamil or English). A Tamil-Sinhala-English translator will be available during the interview for assistance upon permission from the participant. For audio recordings, participants will be advised not to identify themselves or third parties. Digital recordings of the interviews and transcripts will be password protected and saved on a secure Google Drive linked to the researcher's faculty email address. The data will later be transferred to an external storage device once the analysis is completed. Transcribing will be done by the researcher. All transcripts will identify interviewees by codes. Participants will be allowed to review the transcription and provide written consent to use the recording.

**Secondary data:** Policies, news reports, press releases will be analysed to identify any significant events. Online resources will be retrieved using the keywords "Sri Lanka", "Patient safety", "policies", "news reports", and "press releases" in internet search engines.

## Data analysis

**Phase 1: Extensive data.**

**Curriculum content verification:** Percentages will be calculated on the inclusion of five selected concepts in each curriculum. The selected adverse events related to patient safety concepts are, 1) the definition of adverse events (AE) in healthcare, 2) Adverse event reporting, 3) human factors in healthcare safety, 4) system factors in healthcare safety and 5) root cause analysis (RCA).

Students' knowledge outcomes mentioned in WHO-MPSC relevant to the above concepts (total 11) and introduction to incident reporting form provided by the Directorate of Healthcare Quality and Safety, Sri Lanka will be checked against the local curricula of each participating institution (S2 File). Observations will be marked under 3 categories "Not mentioned", "Partially mentioned" and "Explicitly mentioned". Percentages for each category will be calculated in all participating institutes.

**Self-administered questionnaire** Psychometric properties of II scales' Sinhala version will be assessed. Five (5) experts will be consulted to comment on the content validity of the scale. Content Validity Index (CVI) will be calculated to assess the relevancy and clarity of items. Statistical analysis will be conducted using IBM SPSS Statistics version 20.0

for Windows. Reliability will be assessed using measurements of internal consistency (Cronbach's alpha coefficient and Composite reliability). Exploratory factor analysis (EFA) will be conducted to identify the constructs of the instrument using Principal Component Analysis (PCA), Varimax rotation and Kaiser Normalization. Construct validity will be evaluated using convergent and divergent validity and confirmatory factor analysis (CFA). Thereafter, Individual innovativeness categories of nursing educators will be identified using the II scoring matrix (S3. File).

The validity and reliability of the adapted SoCQ Sinhala version will be assessed. SoC quick scoring device (S1 Fig) will be used to calculate the scores of individual nursing educators and determine stages of concern in each. Frequency, mean, SD and Percentiles will also be calculated.

**Phase 3: Linking phase.** Correlations of curricula content, personal characteristics, II scale and SoC will be assessed using SPSS and AMOS. The results will be used to determine how Individual innovativeness and personal characteristics affect SoC. An attempt will be made to determine if the SoC of nursing educators affect the integration of AER- PS content in nursing curricula. Actual events will be identified and re-described using data collected in Phase 1 and the conceptual framework.

**Phase 4: Intensive data.** Thematic analysis will be used to analyse transcriptions of semi-structured interviews. Themes will be identified using NVivo. Deductive thematic analysis will be conducted, based on the conceptual framework presented. Yet, the researchers will not get limited by the framework and will also include inductive analysis, given that new constructs could appear during the analysis phase.

Content analysis will be used to analyse secondary data (policies, news reports, and press releases). Characteristics of the document's content will be described by examining who says what, to whom, and with what effect [75].

## Interpretation of data

To comprehend the factors that impact nursing educators' willingness to learn and teach new PS concepts, both quantitative and qualitative analyses will be triangulated [76]. An attempt will be made at identifying how quantitative correlations (personal characteristics, SoC score, II score, curriculum content) and qualitative themes generated (on perceived attributes of patient safety education, resources, opportunities, information and support) would integrate to provide an informative, layered explanation of why nursing educators' perceptions have become what they are. Retroductive approach and causal analysis [57] will be used to identify individual, environmental and curricula-related causes which could have contributed to the willingness to learn and teach.

## Ethical considerations and declarations

Ethical clearance to conduct this study has been obtained by the Ethical Review Committee at the Faculty of Graduate Studies, University of Colombo (FGS/ERC/2022/013).

The researchers have obtained necessary permissions from the copyright owners to conduct a translation and validation of the Individual Innovativeness scale, as well as to reuse the Stages of Concern questionnaire and its corresponding Sinhala translation.

**Risk-benefit assessment.** There are no interventions that are used during the study which requires special precautions. Therefore, there are no potential risks to the participants. There are no immediate benefits to the participants either. Social and scientific value of the study are mentioned in significance of the study. Participants will be directed to WHO-Multi Professional Patient Safety Curriculum and other patient safety reading materials at the end of the questionnaire.

**Participants rights and consent.** An email and/or message will be sent to all nursing educators who have met inclusion criteria requesting to participate in the study. Participants will be recruited through volunteer participation. A brief description of the study, ethical approval, details of investigator and supervisor will be provided in the information sheet hyperlinked to the questionnaire. In 2nd section in google form, there will be five questions including, "have you read

the information sheet? Do you agree to take part in this study?" If they select YES for all 5 questions and proceed to the questionnaire, it will be considered as consent given. If they select NO for at least one question, they will exit from the google form without proceeding to the questionnaire. In the information sheet, it will be stated that the participants can withdraw their consent to participate in the study at any point, even after submitting the questionnaire or after participating in interviews. Their data will be duly removed from the study, if they opt out. Interview participants had the right to ask questions, to withdraw from the study at any time, to withdraw their data, to decline to answer any question, to turn off the recorder. The information sheet will provide contact details of the investigator and the ethical approval committee to register any complaints on the study. Interview participants will have the right to check transcription summaries for accuracy and to receive a summary of the findings of the completed research.

**Confidentiality and privacy.** Demographic data such as age, designation, university will be requested to be used in finding out associations and correlation. Names of the participants will be requested in order to contact participants for semi-structured interviews. Name of the participant and university will be replaced with codes when processing data to ensure privacy of the participant. If data are requested during publications, anonymity of the participants will be ensured. Security of data will be ensured. Spreadsheets and recordings will be saved in the google drive attached to the researcher's faculty e-mail address provided by University of Sri Jayewardenepura with password protected access only permitted to the researcher. Data will be transported from google drive to an external data storage device post thesis submission in order to free cloud storage space. Data will be saved in password protected documents and will be deleted 10 years after submission of thesis. Only the principal investigator and supervisor will have the access to the data.

**Fair participant selection and vulnerability.** Nursing academics who fulfil the inclusion criteria are selected regardless of ethnicity for the online questionnaire. However, due to the lack of Tamil language proficiency of the researcher, if there will be any interviews with Tamil speaking participants, a Tamil-Sinhala-English translator will be available during the interview for assistance upon permission from the participant. If the Tamil speaking participant is fluent in Sinhala and or English, interview will be conducted in participants preferred language (Sinhala/English).

## Discussion

This study aims to identify the willingness of nursing educators in adopting adverse events related to PS concepts, as well as the factors contributing to their willingness. The study will also explore the impact of individual innovativeness on their concerns. The findings of this study will provide valuable insight into the factors that influence nursing educators' willingness, which can be used to propose changes and improvements to individual, environmental, and PS content. Ultimately, improved education on PS will benefit patients by improving their experiences and outcomes at healthcare facilities.

Previous studies have identified the impact of nursing educators' concerns or perspectives in PS education [77–80], but few have delved into the specific causative reasons. It is important to address any underlying causes of these concerns to ensure full participation and commitment from nursing educators. However, in a complex society where causation is often multifaceted and debatable, it can be challenging to identify the exact causes of a phenomenon. Critical realism acknowledges this complexity and proposes the emergence of actual events from generative mechanisms and structures.

While the primary focus is on Sri Lanka, the study's insights and methodological approach have broader applicability to similar studies worldwide. By understanding the underlying mechanisms shaping nursing education in Sri Lanka, the research aims to provide context-specific suggestions, while also contributing to the broader knowledge base in education research.

The researchers recognize that critical realism is a topic of ongoing debate and that some may have reservations about its practical applications, particularly in the three domains of reality (real, actual and empirical) [81]. Additionally, there is limited research on healthcare education utilizing the staged model for explanatory social sciences or proposed Critical Realist Mixed-Methods Sequential Explanatory Design. Conducting a study that combines healthcare education, critical realism, innovativeness, and stages of concern is a challenging and interdisciplinary undertaking. The researchers are aware of these challenges and will strive to navigate them using the best available scientific reasoning.

The study results will only be generalizable among Sri Lankan nursing educators who are fluent in Sinhala, due to the use of Sinhala translations of study instruments. When a study is conducted in the English language in a country where the population is composed of non-native English speakers, it could significantly alter the outcome [63,72,82,83]. There is a possibility that the level of English proficiency of the participants affects the quality, nuances, and richness of the responses they provide. In a self-administered questionnaire, it could influence the way they perceive questions and their intentions. Therefore, validating the Individual Innovativeness scale in local languages was a necessity before using it. However, due to the lack of Tamil-fluent educators, validating the Tamil translation was not feasible. Hence, only the Sinhala translation will be validated and used in the study.

There is a possibility that those who are not willing to accept a new idea or new PS concepts in particular, will refuse to participate in this study. If so, it won't be possible to identify/synthesis what causes them to deny this new concept.

The study's findings will be disseminated through several channels. Firstly, these will be presented at research conferences and submitted for publication in peer-reviewed scientific journals. Additionally, a summary of the results will be provided to nursing department heads and participating educators. A report of the study's findings will also be shared with the Director of Nursing Education and the Directorate of Healthcare Quality and Patient Safety in Sri Lanka. This will ensure that the results of this study are made available to relevant stakeholders and contribute to the wider conversation around PS education in nursing.

## Supporting information

**S1 File. Interview protocol.**
(DOCX)

**S2 File. Curricula data extraction sheet.**
(DOCX)

**S3 File. Individual innovativeness scale score calculation.**
(DOCX)

**S1 Fig. SoC quick scoring device.**
(TIF)

## Author contributions

**Conceptualization:** D.M.A.P. Dissanayake.

**Methodology:** D.M.A.P. Dissanayake, S.S.P. Warnakulasuriya.

**Supervision:** S.S.P. Warnakulasuriya.

**Writing – original draft:** D.M.A.P. Dissanayake.

**Writing – review & editing:** S.S.P. Warnakulasuriya.

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
