## [Decision Letter · Decision Letter 0]

29 Jun 2023

PONE-D-23-13398A critical realist analysis of nursing educators’ concerns on patient safety education in Sri Lanka: study protocolPLOS ONE

Dear Dr. Dissanayake,

Thank you for submitting your manuscript to PLOS ONE. After careful consideration, we feel that it has merit but does not fully meet PLOS ONE’s publication criteria as it currently stands. Therefore, we invite you to submit a revised version of the manuscript that addresses the points raised during the review process.

We look forward to receiving your revised manuscript.

Kind regards,

Baby Gobin

Academic Editor

PLOS ONE

Reviewers' comments:

Reviewer's Responses to Questions

**Comments to the Author**

1. Does the manuscript provide a valid rationale for the proposed study, with clearly identified and justified research questions?

Reviewer #1: Yes

Reviewer #2: Yes

2. Is the protocol technically sound and planned in a manner that will lead to a meaningful outcome and allow testing the stated hypotheses?

Reviewer #1: Partly

Reviewer #2: Yes

3. Is the methodology feasible and described in sufficient detail to allow the work to be replicable?

Reviewer #1: Yes

Reviewer #2: Yes

4. Have the authors described where all data underlying the findings will be made available when the study is complete?

Reviewer #1: No

Reviewer #2: No

5. Is the manuscript presented in an intelligible fashion and written in standard English?

Reviewer #1: Yes

Reviewer #2: Yes

6. Review Comments to the Author

You may also provide optional suggestions and comments to authors that they might find helpful in planning their study.

Reviewer #1: This study protocol describes a study using critical realistic analysis to examine nursing educators’ concerns on patient safety education in Sri Lanka. The author have given a thorough description of the study background, the study objectives and purposes, and the intended methodology.

While the protocol is well written, I would like to see more discussion on the use of critical realism as the methodology. From what I could gather, the protocol described a mixed methods approach, and it is unclear about the justification/demonstration of the use of critical realism as the research method. Furthermore, it would be useful for the authors to clearly define the limitations of other empirical approaches (p.11, line 18), in order to introduce critical realism.

It would have been useful for the authors to describe their hypotheses (at least for the quantitative study).

Have the study instruments been validated for their use in Sri Lanka or South East Asia? It’d be great to include some validation evidence in the study protocol.

It would have been critical for the authors to justify the sample size and sampling strategy. Has a power analysis been conducted to ensure the sample is adequate?

In the abstract under study design, it would have been useful to describe the study design in a sequential order (i.e., Phase 1, Phase 2, Phase 3, and Phase 4).

Reviewer #2: I have attached review as an attachement.

General Comment

1. Please clarify this,

• As far as I understood, Sri Lankan nursing educators are very competent in English. Also, the medium of instruction is English in most Sri Lankan nursing institutes. In this context, what is necessary to translate the questionnaire into a local language?

• If done so, how can you generalize your result in Sri Lanka since there are two official languages (Sinhala and Tamil)?

2. Please emphasise in the manuscript that you will use two sets of samples separately for the Psychometric study and questionnaire survey.

3. Follow the same style of citation throughout the manuscript. Also, use correct form of citation (full stop follows after bracket – See P6 L2 and P6 L9).

7. PLOS authors have the option to publish the peer review history of their article (what does this mean? ). If published, this will include your full peer review and any attached files.

**Do you want your identity to be public for this peer review?** For information about this choice, including consent withdrawal, please see our Privacy Policy .

Reviewer #1: **Yes: ** Yan Chen

Reviewer #2: **Yes: ** Dr. P. Youhasan

---

## [Author Response · Author response to Decision Letter 1]

9 Aug 2023

Dear Editor and reviewers,

Thank you for reviewing our manuscript. We highly appreciate the time and effort you have made to point out the amendments needed. We have tried our best to provide sufficient rationale to each of our decisions. Please find below, the responses and details of amendments done to the manuscript.

Editor:

Style updated.

2. Your ethics statement should only appear in the Methods section of your manuscript. If your ethics statement is written in any section besides the Methods, please delete it from any other section. Moved to methods section. L 563

3. Please include a separate caption for each figure in your manuscript. Included. L 275, L 337

4. Have the authors described where all data underlying the findings will be made available when the study is complete?

Data will be made available once the study is complete. At this stage researcher does not have any data to be shared.

Availability of data and materials –Data will be available upon request when the study is completed and published. L9-10

Reviewer 1

1. The authors have given a thorough description of the study background, the study objectives and purposes, and the intended methodology.

While the protocol is well written, I would like to see more discussion on the use of critical realism as the methodology. From what I could gather, the protocol described a mixed methods approach, and it is unclear about the justification/demonstration of the use of critical realism as the research method. Thank you for your positive feedback.

We included the definition of “methodology” in use: “Methodology is concerned with the theoretical and philosophical assumptions about methods.(1) It provides the frame of reference for the research which is influenced by the paradigm in which our theoretical perspective is placed or developed…. Methodology can be objectivist, subjectivist or critical realist.”

Methodology- Critical realism

Data collection methods- mixed method L 212-L 218

2.Furthermore, it would be useful for the authors to clearly define the limitations of other empirical approaches (p.11, line 18), in order to introduce critical realism. In this study empirical evidence will be obtained on the stages of concerns, in individual, environmental and innovation related areas, and critical realism’s depth reality will be used to understand complexities that are not obvious and visible. L 272-274 Limitations of objective and subjective methodologies added. L219-L250

3. It would have been useful for the authors to describe their hypotheses (at least for the quantitative study). Included in L 292-296

4. Have the study instruments been validated for their use in Sri Lanka or South East Asia? It’d be great to include some validation evidence in the study protocol. Included in SoCQ validation evidence L425-L433.II scale L 442-443

5.It would have been critical for the authors to justify the sample size and sampling strategy. Has a power analysis been conducted to ensure the sample is adequate? Sampling strategy was altered post reviewer comments. New calculation included. Updated: L 401-405

Total population (updated) =413

Tamil speaking educators= 35 (approx)

Sinhala fluent educators = 378

σ = √(p(1-p)) = 0.5.

α = 1 - 0.95 = 0.05.

p = 1 - α = 1 - 0.05 = 0.975

2 2

n = Z20.975 * p(1 - p)

MOE2

n = 1.962 * 0.5(1 - 0.5) = 384.1459

0.052

Rounded up to: 385.

Since the population size is finite: N=278, the corrected sample size is:

n' = n * N

n + N - 1

n' = 384.1459 * 278 = 161.5265

384.1459 + 278 - 1

Rounded up to: 162.

Using Yamane's formula with a population size of 273 and a margin of error of 5%,

sample size for 2nd sample can be calculated as follows:

n = N / (1 + N(e)*2)

n = 273 / (1 + 273(0.05) *2)

n = 273 / (1 + 273(0.0025))

n = 273 / (1 + 0.6825)

n = 273 / 1.6825

n ≈ 162.29

n=162

Using Yamane's formula (2,3) n = N / (1 + N(e)^2), for a population of 273 with margin of error of 5%, sample is calculated to be 162+ non respondents 5% = 171.

Considering factor analysis for SoC questionnaire (35 items; 1 item: 5 educators), 175 educators are required to calculate CFA. A total of 185 results after adding 5% as non-respondents. The higher value, 185 will be taken as the appropriate sample size for sample 2

6. In the abstract under study design, it would have been useful to describe the study design in a sequential order (i.e., Phase 1, Phase 2, Phase 3, and Phase 4). Corrected. Data collection, analysis and interpretation will take place in four consecutive phases; phase 1, linking phase, phase 2 and interpretation phase L 29-38

Reviewer 2

General Comment

1. Please clarify this,

• As far as I understood, Sri Lankan nursing educators are very competent in English. Also, the medium of instruction is English in most Sri Lankan nursing institutes. In this context, what is necessary to translate the questionnaire into a local language? We opted for a translated instrument due to following reasons:

1. Using a validated tool for a particular population is mandatory an academic research. As Gjersing et al., explain (4)"However, a previously validated instrument does not necessarily mean it is valid in another time, culture or context."

2. If a study is conducted in English language, in a country where the population is non-native English speakers, it could hamper the quality of data. There is a possibility that level of English proficiency of the participant affects the quality, nuances and the richness of responses they provide. In a self-administered questionnaire, it could affect the way they perceive questions and their intensions.

3. Researchers have emphasized the necessity of translation to a local language of the intended population and cultural adaptation of research questionnaires/ tools when validating.(5–7)

If done so, how can you generalize your result in Sri Lanka since there are two official languages (Sinhala and Tamil)? Results will not be able generalize in Sri Lanka. Study results will only be generalized among the Sri Lankan nursing educators who are fluent in Sinhalese Thank you for highlighting this. It will be specified as a limitation. Updated in Discussion. L 601

2. Please emphasise in the manuscript that you will use two sets of samples separately for the Psychometric study and questionnaire survey. For the self-administered questionnaire, two sets (sample 1 and sample 2) of nursing academics will be recruited. Data from sample 2 will be used to questionnaire survey analysis L 379-382

3. Follow the same style of citation throughout the manuscript. Also, use correct form of citation (full stop follows after bracket – See P6 L2 and P6 L9). Corrected.

ABSTRACT

I. P1 L16. College of Nursing or NTS College of Nursing is also a commonly used term. Nurses Training School was used as it is the term used in official documents by the Ministry of Health (NTS -Nurses Training School) 1. latest available Annual health bulletin produced by Ministry of Health, Sri Lanka- 2020, page XXV.

2.http://www.health.gov.lk/moh_final/english/others.php?pid=99

II. P1 L21. DAARCC - What is this abbreviation means? D- description

A -analytical resolution

A- abduction/theoretical redescription

R- retroduction

C-comparison between different theories and abstractions

C-concretization and contextualization

Used in CRMMSED.(8)

Removed abbreviation as it deemed inappropriate for the abstract. Changed the text to "The stages in an explanatory research based on critical realism and Mixed method.."

III. What will happen in phase-04? Wording of phases corrected according to CRMMSED.

It is the interpretation phase where we will compare/triangulate quantitative data analysis and qualitative data analysis.(9)

The aim is to identify how individual, environmental and patient safety education content bring about different stages of concern in educators.

Further, aligning our analysis with the critical realistic depth reality, we will employ critical realist causal analysis techniques (10)to identify events structures and mechanisms that contribute to each stage of concern.

III. What is the rationale for keeping a phase for data analysis? Corrected, CRMMSED phases will be used throughout. Initially there was a mixed up with original MMSE design. In phase 2, qualitative data will be collected through semi-structured interviews with nursing educators and thematically analysed. L35-36

IV. P2 L9-10. What method will be used to analyse qualitative data? Thematic analysis will be used. In phase 2, qualitative data will be collected through semi-structured interviews with nursing educators and thematically analysed. L35-36

What is already 1 known about the topic?

V. P3 L4. Briefly mention those seven categories or write it in a different way added: " ranging from “unconcerned” to “informational”, “personal”, “management”, “consequence” “collaboration” and “refocusing”. " L 57-59

VI. P3 L9. factors which affects or factors which from Corrected; innovation-related factors which form the concerns of nursing educators, L 63

VII. P3 L11. possibility of synergising Corrected L65

Introduction

VIII. P4 L21 Define an abbreviation when using the first time added abbreviation at L75, and changed throughout the manuscript.

IX. P5 L10-11 Reverse the sentence to avoid confusion – …Patient safety concept must be integrated into the curriculum for producing competent nurses. (This is just an idea) Corrected. Nursing educators must be willing to integrate and teach new PS concepts to produce nurses who can champion PS. L 102-103

X. P5 L14 ‘stages of concern’ – Please define the new concept when using the first time in the manuscript. removed stages of concern as it was identified as too early to mention. It will be explained in details under Concerns Based Adoption Model (CBAM) Changed to “Therefore, further research is needed to evaluate both the curricula and nurse educators' concerns on teaching these concepts. L 102-105

XI. P6 L1-3 – Reverse the sentence (Look at the comment for P5 L10-11). rewrote: The lack of resources is the main barrier to promoting patient safety and quality practices in India. L 11-112

XII. P6 L7-9 –Please rewrite. Quite confusing rewrote: A study done in a teaching hospital in Sri Lanka revealed that nursing officers who are competent in skills regard to adverse events and, near misses were 6.35%. L 116-118

XIII. P7 L1-7 – Include more information about the patient safety curriculum – Especially curriculum content Content included: L130-136

XIV. P7 L14 - becoming competent using it – this should correct as ‘becoming competent in using it’ or ‘becoming a competent user’ Changed: becoming competent in using it. L149

XV. P7 L16 – Sate reference for your claim A 35-item Stages of Concern questionnaire is available in order to determine where someone is in the Stages of Concern range.(11) L 155-156

XVI. P8 L5-6 – Rewrite the sentence. Eg. …innovation highly depends on personal choice or innovation is highly influenced by intrinsic factors of a faculty. changed to: it is important to note that acceptance and adoption of an innovation is highly influenced by intrinsic factors of an individual. L 157-158

XVII. P9 L6 - Sate reference for your claim Roger's diffusion of innovation theory. Updated L 175

XVIII. P9 L8 - Sate reference for your claim Roger's diffusion of innovation theory L 178

XIX. P9 L11 – follow a referencing style - Hurt et al (1997) or Hurt et al (33)? Corrected. Hurt et al., (34)L 181

XX. P9 L12 – Replace (ii) by as Hurt et al (1997) define Individual Innovativeness (II) as the “willingness of an individual to change”.L181

XXI. P9 L19 – Replace involves by includes or incorporates changed to: includes L189

XXII. P10 L4 – Relate how the educational environment influences adoption Research done in the management and IT industries has found that positive working environment such as strong managerial support, allocation of sufficient resources and incentives become crucial catalysts in embracing innovation. As such, it could be predicted that nursing educators' environment could affect education innovation adoption. Updated L 196-199

XXIII. P11 L3 – Critical Corrected. L 258

XXIV. P11 L4 - Replace – ‘is more an’ as ‘is more towards’ We wanted to emphasize was that by nature, CR is essentially an ontology, not that it has more tendency to be an ontology.

Aim of the study

XXV. P12 L14 – Aim should be SMART – don’t include details about methods in the aim Corrected L 290- 291

XXVI. P14 L3 - education system or nursing educators? Changed to Sri Lankan nursing educators. L 324

XXVII. P14 L5 - local nursing curricula in Sri Lanka Corrected L 325

XXVIII. P14 L7 – integrators or developers? Developers L 328

XXIX. P14 L14-16 – Repetition | It is not necessary Removed.

XXX. P14 L22 - State reference – ‘extensive methods and intensive methods’ Updated L 343

Sampling Technique

XXXI. P16 L13 – Are you excluding Tamil-speaking academics? If so, provide a rationale 1. Unavailability of a study sample (number of Tamil speaking educators) to conduct a validation study.

2. Time limitation.

We would like to emphasize that we have no intention of excluding Tamil speaking educators or create any racial discrimination. If they recognise themselves as fluent in Sinhala as the secondary or tertiary language, they will be included in the sample. This will be mentioned as a limitation of the study. Permanent nursing educators (N total=413) who are capable of reading, speaking and comprehending Sinhala language (NS≈378)

L 371-372

XXXII. P16 L14 – Why can’t you include faculty who teach basic sciences to nursing students? 1. It is very unlikely that adverse events related patient safety content will be taught in basic sciences (anatomy, physiology and biochemistry).

2. The study aims to analyse perceptions of nursing educators who teach core nursing subjects and who can influence/ contribute to the improvement of core nursing curricula content. teach at-least one nursing subject (clinical nursing subjects, fundamentals of nursing, management in nursing, trends and issues in nursing etc) whom will have the potential to provide feedback on nursing curriculum, will be selected for the sample. L374-5

XXXIII. P19 L20-21 – How will you follow up?

Also, please mention that this is not an anonymous data collection Non-respondents will receive a reminder email on following week and a phone reminder on the 2nd week. L 483

added in phrase

"Data collection is not anonymized. To protect privacy and confidentiality, names and university affiliations will be replaced with codes during data processing. Any data requested during publications will also be anonymized." L 457-459

Phase 3- Intensive/ qualitative procedures (P19)

XXXIV. P20 L1. Semi-structured interviews - Please attach an interview guide (including a pre-determined set of open questions and interview ground rules) in this section. Attached. S1File

· Where will you conduct the interview? changed to: one to one interviews will be conducted in-person or virtually according to participant’s preference. If done in-person, it will be at the participants' academic institution. L 489- L 491

· Which language will be used? Interview will be conducted in participants preferred language (Sinhala, Tamil or English). A Tamil-Sinhala-English translator will be available during the inter

---

## [Decision Letter · Decision Letter 1]

27 Mar 2024

PONE-D-23-13398R1A critical realist analysis of nursing educators’ concerns on patient safety education in Sri Lanka: study protocolPLOS ONE

Dear Dr. Ashoka,

Thank you for submitting your manuscript to PLOS ONE. After careful consideration, we feel that it has merit but does not fully meet PLOS ONE’s publication criteria as it currently stands. Therefore, we invite you to submit a revised version of the manuscript that addresses the points raised during the review process.

Please submit your revised manuscript by. 11 May 2024. If you will need more time than this to complete your revisions, please reply to this message or contact the journal office at plosone@plos.org . Please include the following items when submitting your revised manuscript:

We look forward to receiving your revised manuscript.

Kind regards,

Surangi Jayakody, MBBS, MSc, MD

Academic Editor

PLOS ONE

Journal Requirements:

Reviewers' comments:

Reviewer's Responses to Questions

**Comments to the Author**

1. Does the manuscript provide a valid rationale for the proposed study, with clearly identified and justified research questions?

Reviewer #2: Yes

Reviewer #3: Yes

2. Is the protocol technically sound and planned in a manner that will lead to a meaningful outcome and allow testing the stated hypotheses?

Reviewer #2: Yes

Reviewer #3: Yes

3. Is the methodology feasible and described in sufficient detail to allow the work to be replicable?

Reviewer #2: Yes

Reviewer #3: Yes

4. Have the authors described where all data underlying the findings will be made available when the study is complete?

Reviewer #2: Yes

Reviewer #3: Yes

5. Is the manuscript presented in an intelligible fashion and written in standard English?

Reviewer #2: Yes

Reviewer #3: Yes

6. Review Comments to the Author

You may also provide optional suggestions and comments to authors that they might find helpful in planning their study.

Reviewer #2: Overall, this is a good study!

Referencing: Does the in-text citation come after a full stop | .(1) | or before a full stop | (1). |

Abstract

L36 –Is it inductive or reductive thematic analysis?

What is already 1 known about the topic?

L63- Please carefully read the previous comment: ….factors which from

Introduction

L181 - Hurt et al (1997) define Individual Innovativeness (II) ….. Here, is the (II) necessary?

Suggestion: Hurt et al., (1997) define Individual Innovativeness as the “willingness of an individual to change”.

Reviewer #3: 1. The abstract should adhere to the prescribed word limit. It is imperative to eliminate extraneous elements, such as ethical considerations, from the abstract.

2. In the context of scholarly writing, it is customary to place citations within brackets at the end of a sentence, with the full stop occurring after the closing bracket.

3. The term "programme" should be spelled with the correct spelling, not as "program.

4. Avoid commencing a sentence with a numeral.

5. Definitions of phrases or concepts should be shortened while ensuring that the meaning remains intact.

6. Avoid the use of abbreviations in the objectives.

7. The "significance of the study" and "conceptual framework" components should precede the objectives.

8. The methodologies that will be employed in the study should be delineated under the section dedicated to study methodology, rather than within the introduction.

9. Whenever referring to the Ministry of Health, use the complete designation "Ministry of Health, Sri Lanka.

10. If data collection is not anonymous, it is imperative to provide a more detailed explanation, particularly focusing on aspects such as confidentiality, data quality, and ethical considerations.

11. In the write-up, use the future tense appropriately to convey actions or events that will occur or be completed in the future.

7. PLOS authors have the option to publish the peer review history of their article (what does this mean? ). If published, this will include your full peer review and any attached files.

**Do you want your identity to be public for this peer review?** For information about this choice, including consent withdrawal, please see our Privacy Policy .

Reviewer #2: **Yes: ** Dr. Punithalingam Youhasan

Reviewer #3: **Yes: ** W.D.C.N.Adikaram

---

## [Author Response · Author response to Decision Letter 2]

29 Apr 2024

Dear Editor and reviewers,

Thank you for reviewing our manuscript yet again. We highly appreciate the time and effort you have made to point out the amendments needed. We have tried our best to provide sufficient rationale to each of our decisions. Please find below, the responses and details of amendments done to the manuscript.

Reviewer #2:

Overall, this is a good study! Thank you very much for your positive feedback. It encourages us to strive and do better.

Referencing:

Does the in-text citation come after a full stop | .(1) | or before a full stop | (1). | We were using Vancouver Referencing Style as directed in journals' author guidelines which instructs to have citations number after full-stop.. But we observed the published articles in the journal has in text citations after full-stop. We have changed accordingly. Thank you. In text citations

Abstract

L36 –Is it inductive or reductive thematic analysis? Deductive thematic analysis will be conducted, based on the conceptual framework presented . Yet, the researchers will not get limited by the framework and will also include inductive analysis, given that new constructs could appear during the analysis phase.(L 683-685). We only mentioned as thematic analysis in the abstract due to word limitation.

What is already 1 known about the topic?

L63- Please carefully read the previous comment: ….factors which from: Thank you for highlighting the error and your suggestion. We wanted to identify the factors which contribute to the emergence of different concerns through a critical realist perspective. We have decided to change it as "factors which contribute to the emergence of different concerns". For your consideration please. L56

Introduction

L181 - Hurt et al (1997) define Individual Innovativeness (II) ….. Here, is the (II) necessary?Suggestion: Hurt et al., (1997) define Individual Innovativeness as the “willingness of an individual to change”. Thank you, changed as suggested. L 175

Reviewer #3:

1. The abstract should adhere to the prescribed word limit. It is imperative to eliminate extraneous elements, such as ethical considerations, from the abstract. Thank you for highlighting this. We have removed ethical considerations from abstract.

2. In the context of scholarly writing, it is customary to place citations within brackets at the end of a sentence, with the full stop occurring after the closing bracket. We were using Vancouver Referencing Style as directed in journals' author guidelines which instructs to have citations number after full-stop. But we observed the published articles in the journal has in text citations after full-stop. We have changed accordingly. Thank you.

3. The term "programme" should be spelled with the correct spelling, not as "program. Noted with thanks. Corrected. L 80, L 328,

4. Avoid commencing a sentence with a numeral. Corrected, thanks. L 111- Nearly half 46%

5. Definitions of phrases or concepts should be shortened while ensuring that the meaning remains intact Corrected, thanks. L 175-The general innovativeness of an individual is typically viewed as a psychological construct or individual characteristic. It shapes an individual’s disposition toward newness regardless of the kind of innovation (33)

L 186- Structural empowerment refers to workplace conditions and policies that facilitate four dimensions. These dimensions are access to information, support....

6. Avoid the use of abbreviations in the objectives. noted with thanks L 327- Patient safety

L 333, L 337, L339

7. The "significance of the study" and "conceptual framework" components should precede the objectives. noted with thanks Changed

8. The methodologies that will be employed in the study should be delineated under the section dedicated to study methodology, rather than within the introduction. The article format suggests Introduction, materials and methods, discussion. But I cannot include methodology under methods as I discuss theoretical foundations under this section. It is more suitable to be in Introduction section rather than methods. So I added in a Methodology sub section in Introduction. For your consideration please L 207

9. Whenever referring to the Ministry of Health, use the complete designation "Ministry of Health, Sri Lanka. Thanks, changes were made. L 24, L 373, L 474

10. If data collection is not anonymous, it is imperative to provide a more detailed explanation, particularly focusing on aspects such as confidentiality, data quality, and ethical considerations. Thank you for highlighting this. We have included details of ethical considerations. L 573-625

11. In the write-up, use the future tense appropriately to convey actions or events that will occur or be completed in the future. Corrected, thanks. L 309- this study will be useful for the future

L 55, L58

L421- will be ascertained

---

## [Decision Letter · Decision Letter 2]

23 Jul 2024

PONE-D-23-13398R2A critical realist analysis of nursing educators’ concerns on patient safety education in Sri Lanka: Study protocolPLOS ONE

Dear Dr. Dissanayake,

Thank you for submitting your manuscript to PLOS ONE. After careful consideration, we feel that it has merit but does not fully meet PLOS ONE’s publication criteria as it currently stands. Therefore, we invite you to submit a revised version of the manuscript that addresses the points raised during the review process. Please submit your revised manuscript by Sep 06 2024 11:59PM. If you will need more time than this to complete your revisions, please reply to this message or contact the journal office at plosone@plos.org . Please include the following items when submitting your revised manuscript:

We look forward to receiving your revised manuscript.

Kind regards,

Surangi Jayakody, MBBS, MSc, MD

Academic Editor

PLOS ONE

Journal Requirements:

Reviewers' comments:

Reviewer's Responses to Questions

**Comments to the Author**

1. Does the manuscript provide a valid rationale for the proposed study, with clearly identified and justified research questions?

Reviewer #4: Yes

Reviewer #5: Yes

2. Is the protocol technically sound and planned in a manner that will lead to a meaningful outcome and allow testing the stated hypotheses?

Reviewer #4: Yes

Reviewer #5: Partly

3. Is the methodology feasible and described in sufficient detail to allow the work to be replicable?

Reviewer #4: Yes

Reviewer #5: Yes

4. Have the authors described where all data underlying the findings will be made available when the study is complete?

Reviewer #4: Yes

Reviewer #5: Yes

5. Is the manuscript presented in an intelligible fashion and written in standard English?

Reviewer #4: Yes

Reviewer #5: Yes

6. Review Comments to the Author

You may also provide optional suggestions and comments to authors that they might find helpful in planning their study.

Reviewer #4: thanks for your valuable article. it seems that the needs for conducting this research should be more explained in the introduction.

Reviewer #5: Abstract - It is more clear if authors mention "Methods" after introduction instead of setting and design. Under methods, you could describe all the components (setting, study population, inclusion and exclusion criteria, sample size, sampling method, study instruments..... ). Under data analysis, what are the statistical tests utilized for analysis?

The study protocol is a lengthy one contain a validation, audit and descriptive studies which involve quantitative and qualitative parts. Better if authors can clearly mention as stages or phases under the whole study.

There are some repetitions which could be avoided and leads to a concise protocol.

As this is not a analytical study, how the authors assess the hypothesis mentioned?

In hypothesis 1 - line 341 - repetition of " nursing educators"

Specific objectives 6 and 7 are more or less same.

Specific objective 8 - Is this realistic?

Inclusion and exclusion criteria - Please mention clearly.

Qualitative component - How do you know sample size of 12 - 14 will come to the saturation point?

The sampling method should be more precise and practical for each study component. Eg. How you are going to recruit 105 from each institution mentioned/ how many from each/ sub sampling mechanisms.....

Validation is planned to be done on Hurt-Joseph-Cook individual innovativeness questionnaire. Pls clearly mention what is the validating procedure? Is it construct validity? If so is it factor analysis? EFA? CFA? Both? Then what are the steps? Is it PCA? .......

But SoC questionnaire is already validated in Sri Lanka with good psychometric properties. Then why you are going to perform CFA on it?

How to ensure the validity and reliability of self administered questionnaire which will be used in study in addition to two standard questionnaires?

As mentioned in the supporting materials, scoring of the Individual Innovation scale, > 68 = high; < 64 = low

Then what about the score between 64 - 68?

Anonymity (which is fundamental in obtaining views clearly and successful manner) could be secured if you are going to use a coding system.

7. PLOS authors have the option to publish the peer review history of their article (what does this mean? ). If published, this will include your full peer review and any attached files.

**Do you want your identity to be public for this peer review?** For information about this choice, including consent withdrawal, please see our Privacy Policy .

Reviewer #4: No

Reviewer #5: No

---

## [Author Response · Author response to Decision Letter 3]

6 Sep 2024

Reviewer #4:

Thanks for your valuable article. it seems that the needs for conducting this research should be more explained in the introduction. Thanks for your encouraging feedback. We have expanded “significance of the study”. L298-309

Reviewer #5:

Abstract - It is clearer if authors mention "Methods" after introduction instead of setting and design. Under methods, you could describe all the components (setting, study population, inclusion and exclusion criteria, sample size, sampling method, study instruments....). Thank you for your suggestion. Changed the subtopics. We did not elaborate on the sampling method, and sampling size in the abstract due to the word limit. But these have been explicitly mentioned in the body of the article.

Under data analysis, what are the statistical tests utilized for analysis? Factor analysis- EFA and CFA, associations and correlations of questionnaire data were included in the abstract. L42-44

The study protocol is a lengthy one. contain a validation, audit and descriptive studies which involve quantitative and qualitative parts. Better if authors can clearly mention as stages or phases under the whole study. This information is provided in Table 2. Comparison of study designs and in Abstract. L30-32

L363

There are some repetitions which could be avoided and leads to a concise protocol. We could avoid some repetitions if we rearrange the manuscript according to the Phases (collating sampling, instruments, data collection and analysis together), But then it will deviate from the manuscript guideline of the journal. Nevertheless, minor changes were done to address this.

As this is not an analytical study, how the authors assess the hypothesis mentioned? Thank you for highlighting this. We agree to your opinion that this study does not necessarily need hypotheses to execute it or for its' analysis. removed hypotheses.

In hypothesis 1 - line 341 - repetition of " nursing educators" removed.

Specific objectives 6 and 7 are more or less same.

6. To examine the perceived attributes and environmental factors that facilitate or hinder the adoption of adverse events related patient safety concepts by nursing educators.

7. To identify the events that contribute to the facilitation or hindrance of the adoption of adverse events related patient safety concepts.

Perceived attributes and environmental factors are in the empirical domain. Phenomena in empirical domain are observed and experienced.

Perceived attributes on an innovative concept cannot be equated to actual events.

We accept that certain "opportunities" participants mention have the possibility of being identified as actual events. To overcome this: if a participant mentions certain opportunity (eg. training) was helpful in accepting new PS concepts, we will use this finding in empirical domain. We will further inquire to find out what directives/policies or recommendations were there enabling these opportunities. Latter findings will be identified as Actual Events.

7. To identify the events that contribute to the facilitation or hindrance of the adoption of adverse events related patient safety concepts.

8. To analyse how variations in willingness to learn and teach PS emerge from the underlying generative mechanisms and structures

To improve clarity we amalgamated the objective on "actual domain” with "real domain" as both objectives are operationalized using causal analysis, retroduction and "synthesis".

7. To synthesis the actual events, generative mechanisms and structures that contribute to the facilitation or hindrance of the adoption of adverse events related patient safety concepts.

Specific objective 8 - Is this realistic? In Critical realism, retroduction is the central mode of inference. Retroduction is the way to dive beyond the measurable elements of observations through events to identify the structures, context, and mechanisms that explain the observation.

Our belief in the existence of a mechanism can be based either on our ability to directly observe it (perceptual criteria), with or without tools to do so, or on our ability to observe its effects (causal criteria) (Bhaskar 1975, p. 179).

Retroduction allows researchers to identify a set of plausible candidate causal mechanisms. This enables a researcher to investigate the potential causal mechanisms and the conditions under which certain outcomes will or will not be realised.

Empirical corroboration allows to selection of the mechanism(s) that offers the best explanation (Mukumbang, 2021).These mechanisms belong to different strata of reality: (i) physical, (ii) biological, (iii) psychological, (iv) psychosocial, socio-economic,(v) cultural, and (vii) normative mechanisms (Danermark, 2019).

Some examples of such analysis: Saxena, D. (2019), Wynn Jr, D. E., & Williams, C. K. (2008) 8. To analyse how variations in willingness to learn and teach PS emerge from the underlying generative mechanisms and structures.

new 7. To synthesis the actual events, generative mechanisms and structures that contribute to the adoption of adverse events related patient safety concepts.

There is a possibility that those who are not willing to accept a new idea or new PS concepts in particular will not even be bothered to answer the questionnaire or participate in the study at all. Therefore, we will not be able to identify/synthesis what causes them to deny this new concept. This will be a limitation of this study and will be acknowledged in limitations. L682-684

Inclusion and exclusion criteria - Please mention clearly. Added in Study Design. L385-394

Qualitative component - How do you know sample size of 12 - 14 will come to the saturation point? We agree that it has to be changed to "semi-structured interviews will be conducted with nursing educators until data saturation is reached". L443-444

The sampling method should be more precise and practical for each study component. Eg. How you are going to recruit 105 from each institution mentioned/ how many from each/ sub sampling mechanisms..... Included subsampling methods in sample calculation L410 L 388- all available curricula in selected institutions will be included in content verification using a purposive sampling technique

L 396-The participants will be selected from three types of institutions (state universities, private universities, and training schools) using stratified random sampling.

L400- participants who were selected using stratified random sampling.

L431- Participants will be selected purposive sampling method.

Validation is planned to be done on Hurt-Joseph-Cook individual innovativeness questionnaire. Pls clearly mention what is the validating procedure? Is it construct validity? If so is it factor analysis? EFA? CFA? Both? Then what are the steps? Is it PCA? .......

EFA and CFA. The steps of validating procedure is mentioned in "Data analysis" Phase 1. L552- L556. Extraction Method: Principal Component Analysis.

Rotation Method: Varimax with Kaiser Normalization.

But SoC questionnaire is already validated in Sri Lanka with good psychometric properties. Then why you are going to perform CFA on it? It is an attempt follow a good practice since wording of the questionnaire was changed to suit patient safety education context. "Subsequent validation" is suggested by Tsang et al. (2017)

How to ensure the validity and reliability of self-administered questionnaire which will be used in study in addition to two standard questionnaires? The self-administered questionnaire has 3 parts.

Part1- Demographics- standard questions are being used.

Part2-SoC questionnaire- cronbatch alpha, subsequent validation CFA

Part3-II questionnaire- cronbatch alpha, validation using EFA and CFA

As mentioned in the supporting materials, scoring of the Individual Innovation scale, > 68 = high; < 64 = low. Then what about the score between 64 - 68? Thank you for pointing this out. We accept that this information is incomplete. Unfortunately, there is no description for the score between 68-64 in the scale. We would like to remove this part as the scoring above this description provides more clarity on innovativeness categories.

Anonymity (which is fundamental in obtaining views clearly and successful manner) could be secured if you are going to use a coding system. Thank you for your valuble suggestion. It is mentioned in the protocol in "Data Collection" L481-482 Data collection is not anonymised. In order to protect privacy and confidentiality; names and university affiliations will be replaced with codes during data processing.

References

Danermark B, Ekström M, Karlsson JC. Explaining society: Critical realism in the social sciences. Routledge; 2019 Mar 20.

Saxena D. Guidelines for conducting a critical realist case study. International Journal of Adult Education and Technology (IJAET). 2021 Apr 1;12(2):18-30.

Tsang S, Royse CF, Terkawi AS. Guidelines for developing, translating, and validating a questionnaire in perioperative and pain medicine. Saudi journal of anaesthesia. 2017 May 1;11(Suppl 1):S80-9.

Wynn Jr DE, Williams CK. Critical realm-based explanatory case study research in information systems.

---

## [Decision Letter · Decision Letter 3]

19 Mar 2025

PONE-D-23-13398R3A critical realist analysis of nursing educators’ willingness to learn and teach patient safety in Sri Lanka: Study protocolPLOS ONE

Dear Dr. Dissanayake,

Thank you for submitting your manuscript to PLOS ONE. After careful consideration, we feel that it has merit but does not fully meet PLOS ONE’s publication criteria as it currently stands. Therefore, we invite you to submit a revised version of the manuscript that addresses the points raised during the review process.The study protocol is lengthy and should be more concise. Exclusion criteria should not simply invert inclusion criteria (e.g., if only Sinhala-conversant participants are included, there's no need to state non-Sinhala speakers are excluded). Also, clarify the need for a translator in lines 686–687 if all participants are Sinhala-conversant. The abstract is too long—consider using subheadings (Background, Methods, Discussion) and summarizing the methods and significance. The introduction is strong, but page 6, line 106, and page 17, lines 353–355, need references. On page 15, line 297, key points should be integrated into the abstract’s discussion.  Please submit your revised manuscript by May 03 2025 11:59PM. If you will need more time than this to complete your revisions, please reply to this message or contact the journal office at plosone@plos.org . Please include the following items when submitting your revised manuscript: 

We look forward to receiving your revised manuscript.

Kind regards,

Musa Adekunle Ayanwale

Academic Editor

PLOS ONE

Journal Requirements:

Reviewers' comments:

Reviewer's Responses to Questions

**Comments to the Author**

1. Does the manuscript provide a valid rationale for the proposed study, with clearly identified and justified research questions?

Reviewer #5: Yes

Reviewer #6: Yes

2. Is the protocol technically sound and planned in a manner that will lead to a meaningful outcome and allow testing the stated hypotheses?

Reviewer #5: Yes

Reviewer #6: Yes

3. Is the methodology feasible and described in sufficient detail to allow the work to be replicable?

Reviewer #5: Yes

Reviewer #6: Yes

4. Have the authors described where all data underlying the findings will be made available when the study is complete?

Reviewer #5: Yes

Reviewer #6: Yes

5. Is the manuscript presented in an intelligible fashion and written in standard English?

Reviewer #5: Yes

Reviewer #6: Yes

6. Review Comments to the Author

You may also provide optional suggestions and comments to authors that they might find helpful in planning their study.

Reviewer #5: The authors have addressed most of the comments to the point.

The study is interesting however, this study protocol is a lengthy. Better if could concise more.

Exclusion criteria should not be the vise versa of the inclusion criteria Eg. In this protocol inclusion criteria will be Sinhala conversant people. Then no need to mention that other than Sinhala conversant people will be excluded from the study.

In line 686/ 687, it is mentioned that there will be a translator???? If the study will include only Sinhala conversant participants, why the investigators will use a translator?

Reviewer #6: Thank you for allowing me to review this study.

The study is generally well written, However I have suggested some minor changes as per below.

-The abstract is too long.

Rather use headings - Background;/Introduction; Methods and analysis and then discussions.

Summarize the methods section and in the discussion briefly provide the significance of your study.

-The introduction is generally well written with relevant literature. Well done to the authors!

-Page 6 line 106 Include references of these studies... An assertion such as this one should have been drawn from some existing information.

Page 15 line 297: Suggestion: A summary of this section of your manuscript and the discussion section at the end of your manuscript is what should be included in the discussion section of your abstract.

Page 16 line 319: These objectives align well with the topic. The study's realist approach ensures deep exploitation of factors, mechanisms and structures influencing nursing educators' willingness to learn and teach patient safety which directly connects to the objectives.

-Page 17 line 353-355 Include the reference for the stat. ement

7. PLOS authors have the option to publish the peer review history of their article (what does this mean? ). If published, this will include your full peer review and any attached files.

**Do you want your identity to be public for this peer review?** For information about this choice, including consent withdrawal, please see our Privacy Policy .

Reviewer #5: No

Reviewer #6: No

---

## [Author Response · Author response to Decision Letter 4]

25 Mar 2025

Dear Editor and reviewers,

Thank you for reviewing our manuscript yet again. We highly appreciate the time and effort you have made to point out the amendments needed. We have tried our best to provide sufficient rationale to each of our decisions. Please find below, the responses and details of amendments done to the manuscript.

Reviewer #5:

The authors have addressed most of the comments to the point. Thank you.

The study is interesting however, this study protocol is a lengthy. Better if could concise more. Thank you for your feedback. We have made some adjustments to the abstract to make it more concise. However, given the broad scope of the study, which covers seven objectives, it is challenging to significantly reduce the content without compromising the clarity and detail necessary to convey the study's comprehensive nature.

Exclusion criteria should not be the vise versa of the inclusion criteria Eg. In this protocol inclusion criteria will be Sinhala conversant people. Then no need to mention that other than Sinhala conversant people will be excluded from the study.

Thank you for highlighting this. We have amended as Academic staff members who are on maternity leave, overseas study leave and overseas sabbatical leave, due to the due to the potential difficulty in including these individuals in semi-structured interviews during the study period, given their absence from the academic setting.

P19, L 374: Academic staff members who are on maternity leave, overseas study leave and overseas sabbatical leave.

In line 686/ 687, it is mentioned that there will be a translator???? If the study will include only Sinhala conversant participants, why the investigators will use a translator?

Thank you for your comment and for raising this important point. While the primary inclusion criterion is that participants should be conversant in Sinhala, we are mindful that there could be variation in language proficiency among the participants. In particular, some Sinhala-speaking participants may also be fluent in Tamil or English (their mother-tongue), and we want to ensure that we accommodate their preferences in terms of the language of the interview. The mention of a Tamil-Sinhala-English translator is to ensure that any potential participants, regardless of their language preferences or proficiency, are provided with the necessary support for clear communication during the interview. The presence of a translator is intended as a precautionary measure and will only be used upon the participant’s permission, ensuring that the process remains flexible and respectful of their language needs.

Reviewer #6:

The study is generally well written; However, I have suggested some minor changes as per below. Thank you.

Rather use headings - Background;/Introduction; Methods and analysis and then discussions.

Summarize the methods section and in the discussion briefly provide the significance of your study. Amended P2, L21-38

-The introduction is generally well written with relevant literature. Well done to the authors! Thank you for your encouraging words.

Page 6 line 106 Include references of these studies... An assertion such as this one should have been drawn from some existing information. Included. P5, L98

Page 15 line 297: Suggestion: A summary of this section of your manuscript and the discussion section at the end of your manuscript is what should be included in the discussion section of your abstract. Amended P2, L31-38

Page 16 line 319: These objectives align well with the topic. The study's realist approach ensures deep exploitation of factors, mechanisms and structures influencing nursing educators' willingness to learn and teach patient safety which directly connects to the objectives. Thank you for validating our approach and confirming the alignment of our objectives with the topic.

Page 17 line 353-355: Include the reference for the statement Included P17-347

---

## [Editor Report · Decision Letter 4]

10 Apr 2025

A critical realist analysis of nursing educators’ willingness to learn and teach patient safety in Sri Lanka: Study protocol

PONE-D-23-13398R4

Dear Dr. Ashoka,

We’re pleased to inform you that your manuscript has been judged scientifically suitable for publication and will be formally accepted for publication once it meets all outstanding technical requirements.

Kind regards,

Musa Adekunle Ayanwale

Academic Editor

PLOS ONE

Additional Editor Comments (optional):

I have reviewed the revised manuscript and the authors’ detailed responses to the reviewers’ comments. It is evident that the authors have thoroughly addressed all raised queries, incorporating the suggested revisions to improve clarity, coherence, and methodological transparency. The updated manuscript meets the journal’s standards and adequately reflects the scope and rigour required for publication. Thank you.
---

## [Editor Report · Acceptance letter]

PONE-D-23-13398R4

PLOS ONE

Dear Dr. Dissanayake,

I'm pleased to inform you that your manuscript has been deemed suitable for publication in PLOS ONE. Congratulations! Your manuscript is now being handed over to our production team.

Kind regards,

on behalf of

Dr. Musa Adekunle Ayanwale

Academic Editor

PLOS ONE